# Agricultural Harvesting Robot Concept Design and System Components: A Review

**Mohd Fazly Mail** [1] 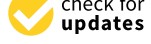, **Joe Mari Maja** [1,*], **Michael Marshall** [2] , **Matthew Cutulle** [2], **Gilbert Miller** [3] **and Edward Barnes** [4]

1   Department of Agricultural Science, Clemson University, 240 McAdams Hall, Clemson, SC 29634, USA; mmail@g.clemson.edu
2   Department of Plant and Environmental Sciences, Clemson University, 171 Poole Agricultural Center, Clemson, SC 29634, USA; marsha3@clemson.edu (M.M.); mcutull@clemson.edu (M.C.)
3   Edisto Research and Education Center, Clemson University, 64 Research Road, Blackville, SC 29817, USA; gmllr@clemson.edu
4   Cotton Incorporated, 6399 Weston Parkway, Cary, NC 27513, USA; ebarnes@cottoninc.com
*   Correspondence: jmaja@clemson.edu

**Abstract:** Developing different robotic platforms for farm operations is vital to addressing the increasing world population. A harvesting robot significantly increases a farm's productivity while farmers focus on other relevant farm operations. From the literature, it could be summarized that the design concepts of the harvesting mechanisms were categorized as grasping and cutting, vacuum suction plucking systems, twisting and plucking mechanisms, and shaking and catching. Meanwhile, robotic system components include the mobile platform, manipulators, and end effectors, sensing and localization, and path planning and navigation. The robotic system must be cost-effective and safe. The findings of this research could contribute to the design process of developing a harvesting robot or developing a harvesting module that can be retrofitted to a commercially available mobile platform. This paper provides an overview of the most recent harvesting robots' different concept designs and system components. In particular, this paper will highlight different agricultural ground mobile platforms and their associated mechanical design, principles, challenges, and limitations to characterize the crop environment relevant to robotic harvesting and to formulate directions for future research and development for cotton harvesting platforms.

**Keywords:** unmanned ground vehicle (UGV); harvesting robot; design concept; harvesting mechanism



## 1. Introduction

Member states of the United Nations adopted the second set of Sustainable Development Goals (SDG) in 2015 to end hunger and ensure access to safe, nutritious, and sufficient food all year round for all people, particularly those in poor and vulnerable situations, including infants. However, there are many challenges to achieving the vision of SDG 2—Zero Hunger by 2030 [1]. The 50 × 2030 Initiative has acknowledged the requirement for precise and up-to-date agricultural and rural data, which could assist in the evaluation and creation of policies aimed at achieving zero hunger and other sustainable development objectives in developing and less-developed nations [2].

The use of technology in achieving sustainable social, environmental, and economic development has become increasingly important through Industrial Revolution 4.0 (IR 4.0). IR 4.0 offers potential solutions to challenges and can contribute to improving quality of life by promoting sustainable products and services. Improvements in technologies that produce alternative methods, techniques, and end products are required to accelerate developments in all research areas [3]. The creation of innovative technology and breakthrough

solutions is essential to achieving the SDG targets, such as poverty eradication, improving food security, and reversing climate change. The integration of autonomous systems generated by machines and data networks has interconnected entire value chains [4].

The research regarding agricultural robots and intelligent machines was reported more than three decades ago [5]. Agricultural robotic systems are applied in various field operations: land preparation before planting, sowing/planting, weeding, seeding, plant treatment, disease detection, crop scouting, spraying, harvesting, and yield estimation and phenotyping for different farming environments [6,7]. This is because agricultural robotics addresses critical issues such as seasonal labor shortages and the growing concern for environmentally friendly practices. The agriculture industry offers poor pay, long work hours, and little chance for advancement. Older workers are leaving the industry, while newer generations show no interest in taking their place. Due to a lack of workers, operations have been delayed and sometimes abandoned. Thus, robotic systems have been designed to cover labor shortages, increase speed, and improve agricultural operation efficiency. The agricultural industry has seen the introduction of agricultural robots, which have proven capable of meeting the growing demand for food by automating farming tasks that were previously labor intensive [7].

### 1.1. Unmanned Ground Vehicle (UGV)

UGV is next-generation technology for farming. Unmanned systems are equipped with wheels, mechanical legs, tracks, propellers, and other external devices that enable interaction with the operational environment and serve as locomotion systems [8]. UGV is widely used in the military at the beginning of deployment to execute dangerous missions or to reach inaccessible places [9]. Autonomously controlled tractors, using commercially available global positioning systems (GPS), and local/global sensors for use in row crops and orchards are already mature [10]. UGV robotic systems are equipped with a range of sensors, including light detection and ranging (LIDAR), red–green–blue (RGB) cameras, time-of-flight (TOF) cameras, near-infrared (NIR) cameras, and stereo vision cameras, to facilitate data collection [11]. In addition, UGV robots are increasingly being converted from internal combustion engines to electric motors.

This review focuses on robots used in harvesting, as they are better suited to agricultural tasks that traditionally require human intervention [12]. Although mechanical harvesting has been around for many years, its impact on soil has come into focus due to erosion and compaction, as most of these machines are heavy. Harvesting is a significant operation, demanding, and challenging area for agricultural robotics. This has been the attention of agricultural experts for many years because of its diversity and complexity, and oftentimes requires different implements for different crops [6,13–15].

### 1.2. Harvesting Method

Manual harvesting involves using bare hands and, in some cases, cutting tools to remove leaves and branches from plants, to hold the fruit, and to pull it out of the plant. Manual harvesting is effective but only if performed by someone with experience, as a novice might accidentally damage the plants. On the other hand, humans have innate gripping ability thanks to their kinematics, sense of touch, and muscular power. They can quickly adjust to the form and texture of crops to apply the appropriate detaching force. However, human capabilities are constrained by exhaustion. Contrarily, harvesting robots can work dependably around the clock. Research has been conducted to create kinematic models for the motion of robotic arms and to build complicated end effectors with adequate sensors to manipulate crops [16,17].

Several excellent reviews of robotic harvesting end effectors have recently been published. Morar et al. presented their robotic end effector development analysis to different crops, e.g., apples, tomatoes, sweet peppers, and cucumbers [16]. An improved manipulator and end effector technology was presented by Davidson et al. [14]. Navas et al. presented recent advances in the design and use of soft grippers for agricultural

harvesting [18], while Vrochidou et al. used modern technologies as a reference to help choose appropriate end effectors for harvesting robots [19,20]. Zhang et al. provided an overview of robotic grippers, gripping methods, sensor-based control approaches, and their use in agriculture and food applications [21].

This review paper covers the most recent harvesting robots' concept design and system components. This paper reviews various studies on mechanical design principles, challenges, and limitations in characterizing the plant environment in the context of robotic harvesting and identifies possible directions for future research and development for cotton harvesting platforms. The main goal is to evaluate existing methods and concepts that could be used to develop an autonomous cotton harvesting robot. The design concepts of the harvesting methods are outlined in Section 2, while the robotic system components are described in detail in Section 3. Finally, Section 4 discusses the results and suggests future work.

## 2. Harvesting Robot Concept Design

Harvesting is an essential process in crop production. While crops such as wheat, soybean, and corn have developed an established machinery system for harvesting, fruity horticultural crops such as apples, tomatoes, grapes, etc., are still manually picked. Contemporary research has narrated various in-depth mechanisms of harvesting fruity crops. It could be summarized that it requires a combination of techniques to pluck the fruit from its stem. The most common combination technique is grasping and cutting, vacuum suction and plucking, twisting and pulling, and shaking and collecting, as categorized in Figure 1.

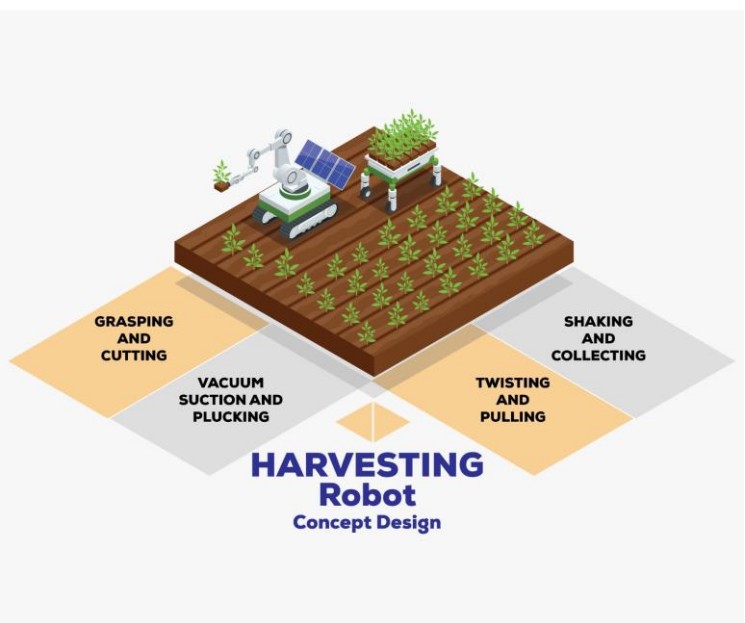

**Figure 1.** Harvesting robot concept design category.

### 2.1. Grasping and Cutting

Grasping and cutting could be applied to crops such as tomato, strawberry, sweet pepper, lettuce, eggplant, asparagus, citrus, grape, and pumpkin. One example is developing a cooperative two-arm robot harvesting process for tomatoes using an open-loop control system. The control system consists of five steps: scanning, detecting the tomato, 3D reconstruction of the scene, gripping with the right arm, and picking with the left arm. The robot is equipped with two different end effectors attached to the left and right arms: a cutting gripper and a suction cup, respectively. In the two-arm manipulator, the vacuum suction cup holds the target crop while the cutting gripper separates it from the plant. The stalk is separated with a double cutter, and the attached grippers are activated to secure

or release the stalk. The drive cylinder is responsible for the rotation of the active blade (Figure 2a). After evaluating the mechanical properties, the CD55B20-30 SMC cylinder with a minimum drive air pressure of 0.3 MPa was selected [22].

As for the strawberries, the grabber has an internal bin that allows it to continuously pick multiple strawberries until the bin is full, reducing the time required for this task. When the bin is full, the strawberries are emptied into another storage bin through a trapdoor at the bottom of the grab. Depending on the size of the strawberries, 7 to 12 berries can be stored in the bin, and the berries are dropped onto an inclined board to minimize impact during the fall. There are also soft sponges inside the gripper to prevent the berries from being damaged by the impact. To achieve optimal control of the cutting position, it is important to know the position of the strawberry in relation to the gripper. The gripper consists of three active and three passive fingers that move simultaneously. Despite their rotation, the lid fingers remain connected to the active fingers via small tension springs. After closing the fingers, a knife with two curved blades (Ideal Tek, Switzerland) rotates quickly to cut the stem and it is hidden in the fingers so that the strawberries are not cut [23]. Compared to the work of Yamamoto et al. [24], a gripper with two fingers was used for cutting the peduncle, and a reflection-type of photoelectric sensor was added to verify the presence of the selected fruit made up of the end effector. To allow for simultaneous gripping and cutting of the handle, a padding material was attached to the contact side of one of the fingers, which was connected to a replaceable blade and stopper [25].

Sweet pepper harvesting uses a mechanism to fix the stem of the plant and a vibrating knife to cut the fruit stem (Figure 2b). The blade is located above the fruit stem and is protected by the fixing mechanism to avoid damaging the plant. When the cutting operation is initiated, the fixing mechanism is raised to provide the blade space to cut through the stem. The mechanism for grasping the fruit consists of six metal fingers covered with soft plastic and is attached to the top of the end effector's camera body. The fingers are spring-loaded so that they can rotate independently when they encounter obstacles as the end effector approaches. Once the stem is cut, the fingers hold onto the fruit [26,27]. Lehnert et al. [27] developed two different methods to determine the grasping and cutting position of sweet peppers. In the first method, a 3D point cloud of the segmented sweet peppers was used to determine the gripping positions for each fruit. In the gripping position, the suction cup was placed directly on the apartment surface of the sweet pepper. The gripping position was calculated to be in the center of the front of the sweet pepper, while the cutting position was calculated to be offset from the top of the fruit. The second method used a different strategy by selecting multiple grip positions directly from the point cloud data, using a specific patch size to calculate the surface normal. The pose was rotated along its axis to maintain the end effector and minimize significant changes in wrist configuration [27].

Birrell et al. [28] developed a lettuce harvesting robot with an end effector containing two pneumatic actuators—one for grasping and one for cutting to allow easy control. The end effector is designed to efficiently cut the lettuce stems while holding the lettuce head to prevent damage. To ensure smooth movement, the linear motion of a single actuator is transmitted to both sides of the blade using a toothed belt system. The actuator can be mounted higher than the height of the lettuce so as not to interfere with cutting, and the height of the cutting mechanism can be easily adjusted with the belt drive system. However, the weight of the end effector causes the path section on the robot arm to slow down considerably [28].

Sepulveda et al. [29] conducted a study on robotic eggplant harvesting and found that the most effective method for leaf movement was to move the arms parallel to the y-axis with a closed gripper. In this way, the device simplifies the movement and avoids using the gripper to grab the leaves. Once the leaves are displaced, the device determines a new center of gravity for the entire eggplant to accurately trim the stem and protect the vegetable. This method is recommended for all capping situations, as the point of

contact with the eggplant remains the same regardless of the distance of the leaf block. In this method, only the leaves are displaced, and the eggplant is not manipulated, so no damage is done. The experiments were carried out with a two-armed robotic platform under laboratory conditions [29].

Designed for asparagus harvesting, the robotic arm offers tilt adjustment of up to 15 degrees to accommodate different growing fields. At the end of the arm are two fingers and a cutting blade. One of the advantages of this system is the use of a cylindrical cam mechanism, which allows for fast operation [30].

Researchers developed a robotic end effector for harvesting citrus, inspired by the head mechanism of snakes, and has a bite mode. The end effector can grasp the citrus fruit and cut off its stem with a scissor-like cutting device, similar to a snake's jaw. This approach could promote the development of a more advanced robot for citrus harvesting and optimized harvesting husbandry [31].

For grapes, a special end effector has been developed that combines a robotic gripper, scissors, and 3D-printed fingers to gently cut and hold grapes without damaging them, facilitating both harvesting and green harvesting [19].

The end effector for pumpkin harvesting is a combination machine suitable for various pumpkin sizes and shapes found in natural farms. The end effector has five fingers, each of which consists of seven components that allow for gripping of the pumpkin (Figure 2c). The inner surfaces of each component are provided with rubber volumes in the form of rollers and stabilizers that facilitate the connection, prevent damage to the crop, and create space between the blades and the surface of the pumpkin. It has been found that a 60° angled blade cuts through the stem with less force and in less time, allowing the sharp blade to quickly cut through the stem by rotating the end effector after gripping the pumpkin [32]. Table 1 provides a summary of the different grasping and cutting methods utilized in robotic harvesting.

**Table 1.** Summary comparison table for the different grasping and cutting methods used in robotic harvesting.

| Crop | End Effector Types | Grasping Mechanism | Cutting Mechanism | Special Features | Cited Work |
|---|---|---|---|---|---|
| Tomato | Vacuum cup and cutting gripper | Dual-arm manipulator | Double cutter and grasping attachment | Open-loop control system, 3D scene reconstruction | [22] |
| Strawberry | Gripper with internal container and three active fingers | Three active and three passive fingers | Two curved blades | Inclined dropping board, soft sponge fitted inside gripper, cover fingers | [23–25] |
| Sweet pepper | Six metal fingers covered in soft plastic | Spring-loaded fingers to rotate around | Plant stem fixing mechanism with vibrating blade | Grasp positions identified from segmented 3D point cloud; several grasping poses selected from point cloud data | [26,27] |
| Lettuce | N/A | Two pneumatic actuators for grasping and cutting | Blade with timing belt system | Linear action transferred to both sides of the blade | [28] |
| Eggplant | N/A | Arms parallel to y-axis with gripper closed | N/A | Avoid using gripper to grab leaves, decreases complexity | [29] |

**Table 1.** *Cont.*

| Crop | End Effector Types | Grasping Mechanism | Cutting Mechanism | Special Features | Cited Work |
|------|-------------------|--------------------|--------------------|-----------------|------------|
| Asparagus | Robotic arm with two fingers and a cutting blade | Cylindrical cam mechanism allows for quick arm movement | Cutting blade | Tilt adjustment function allows the arm to be tilted up to 15 degrees for cultivation field | [30] |
| Citrus | Snake-like end effector with a biting mode | Modeled after the head mechanism of a snake | Scissors | Optimized harvesting postures inspired by bionic principles for a next-generation harvesting robot | [31] |
| Grape | Robotic gripper, scissors, and 3D-printed fingers | 3D-printed fingers with rubber-coated inside surface for defoliation | Scissors | Wide range of finger diameters (76.2–265 mm) for accommodating various grape sizes and shapes | [19] |
| Pumpkin | Five-fingered end effector with seven distinct mechanisms | Each finger has rollers and stabilizers to avoid crop damage | 60° cutting blade | Can accommodate different pumpkin sizes and shapes, sharp blade cuts stem with less force and time, and rotating end effector helps cut stem | [32] |

## 2.2. Vacuum Suction and Plucking

Vacuum suction and plucking methods have been tested in apple and cotton harvesting. Zhang et al. [33] suggested using a vacuum-based end effector for apple harvesting because suction can be effective within a certain area with an appropriate vacuum flow. The researchers demonstrated that the current robotic system could operate successfully with an end effector diameter of 0.04 m. A soft silicone suction cup with a diameter of 0.064 m is attached to the front end of the end effector (Figure 3a). According to the test laboratory, the end effector can adapt to different apple shapes and reduce damage to the fruit owing to the shape of the vacuum cup and the silicone material. To facilitate the detachment of apples, there is a rotating mechanism at the back of the end effector tube. When the manipulator reaches the appropriate apple position, the rotation mechanism is triggered to rotate the entire tube by a certain angle before the manipulator pulls out the apple. The combination of a rotating movement with a pulling movement is more effective than a pure pulling movement. The rear end of the effector tube is connected to a Craftsman electrically operated wet/dry vacuum cleaner via a flexible and expandable hose. Studies have shown that the vacuum-based end effector successfully reduces bruising during harvesting [34]. In another case, a pneumatic gripper with two fingers is used to grasp the peduncle, while a vacuum suction cup holds the fruit, and an electric heat cutter separates it from the plant [35].

A Clemson University research group developed a cotton harvesting robot with the one degree of freedom (DOF) cartesian manipulator that holds a vacuum suction end effector but uses a small rover traversing between the rows (Figure 3b). Before starting to harvest operations, the nozzle's position was set to correspond with the location of most of the cotton bolls along the plant row. However, unsatisfactory performance was observed when the prototype was tested on the cotton crop. In the previous design, there were some problems with the suction distance of the bolts/locks, which changed when the mobile platform moved [36]. To solve this problem, a new and improved harvester was developed. The new design includes a scraper mechanism on the side of the mobile

platform that replaces the suction cap with a rolling scraper. The same suction motor is used to convey the harvested bolls to the bucket. A single 24 V motor drives the stripper on the side. Research is now being conducted to see if the harvesting robot might be used as a once-over harvester [37].

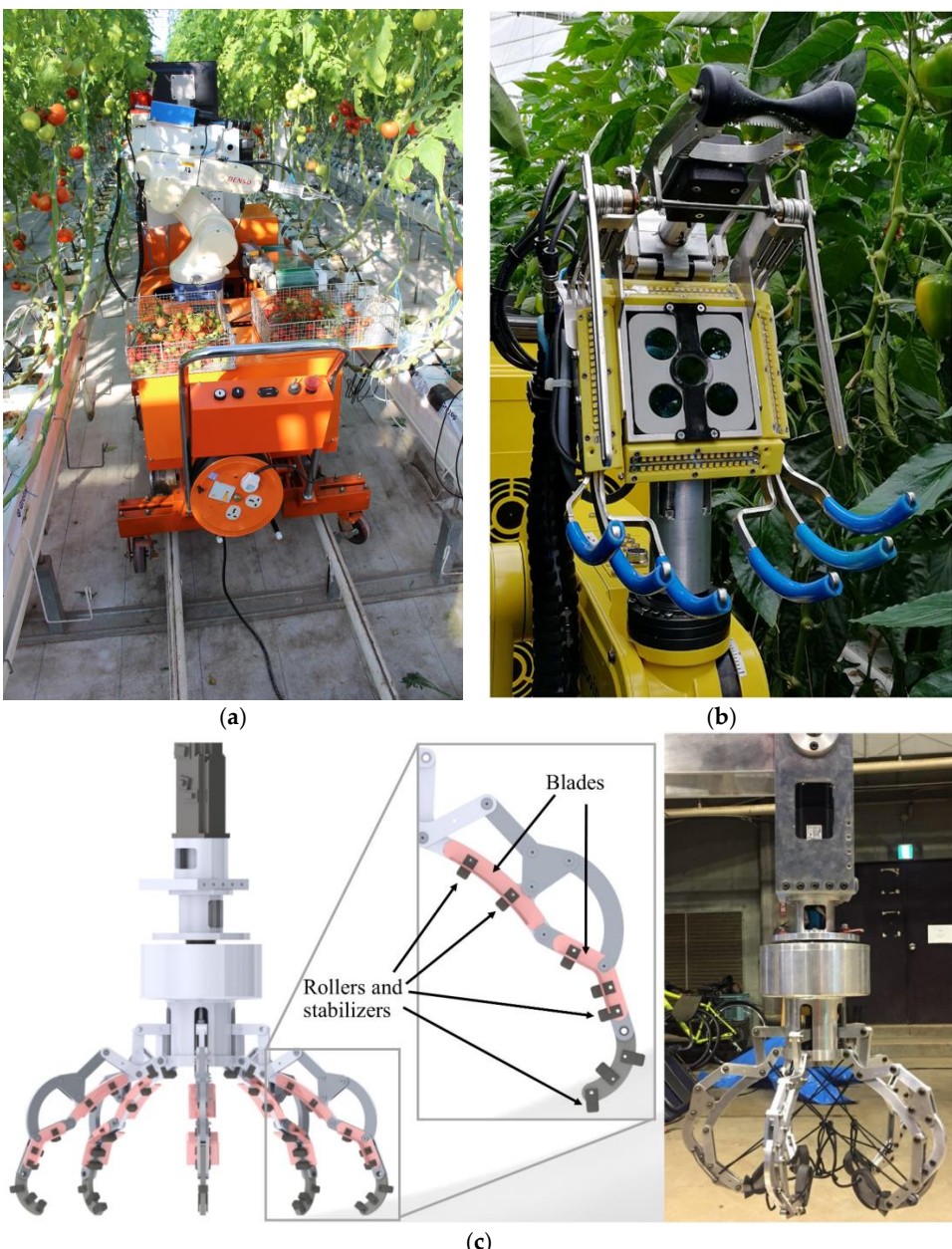

**Figure 2.** Some of the agricultural robots with grasping and cutting concepts: (**a**) robotic harvesting system for cherry tomato [22]; (**b**) sweet pepper harvesting robot [26]; (**c**) pumpkin harvesting robotic end effector [32].

Gharakhani and Thomasson presented a robotic end effector concept for cotton harvesting that was tested and evaluated (Figure 3c). After reviewing various end effector concepts, they selected a configuration with three fingers moving on a needle belt and developed a prototype that met all design criteria, including picking ratio, transfer, deposit, entry and exit, targeting, picking at different angles, and clean and complete picking. During testing, there were failures to pick laterally aligned bolls due to calyx consumption. The control method increased picking time, which could be further optimized to reduce the time required to remove cotton seed from a boll [38].

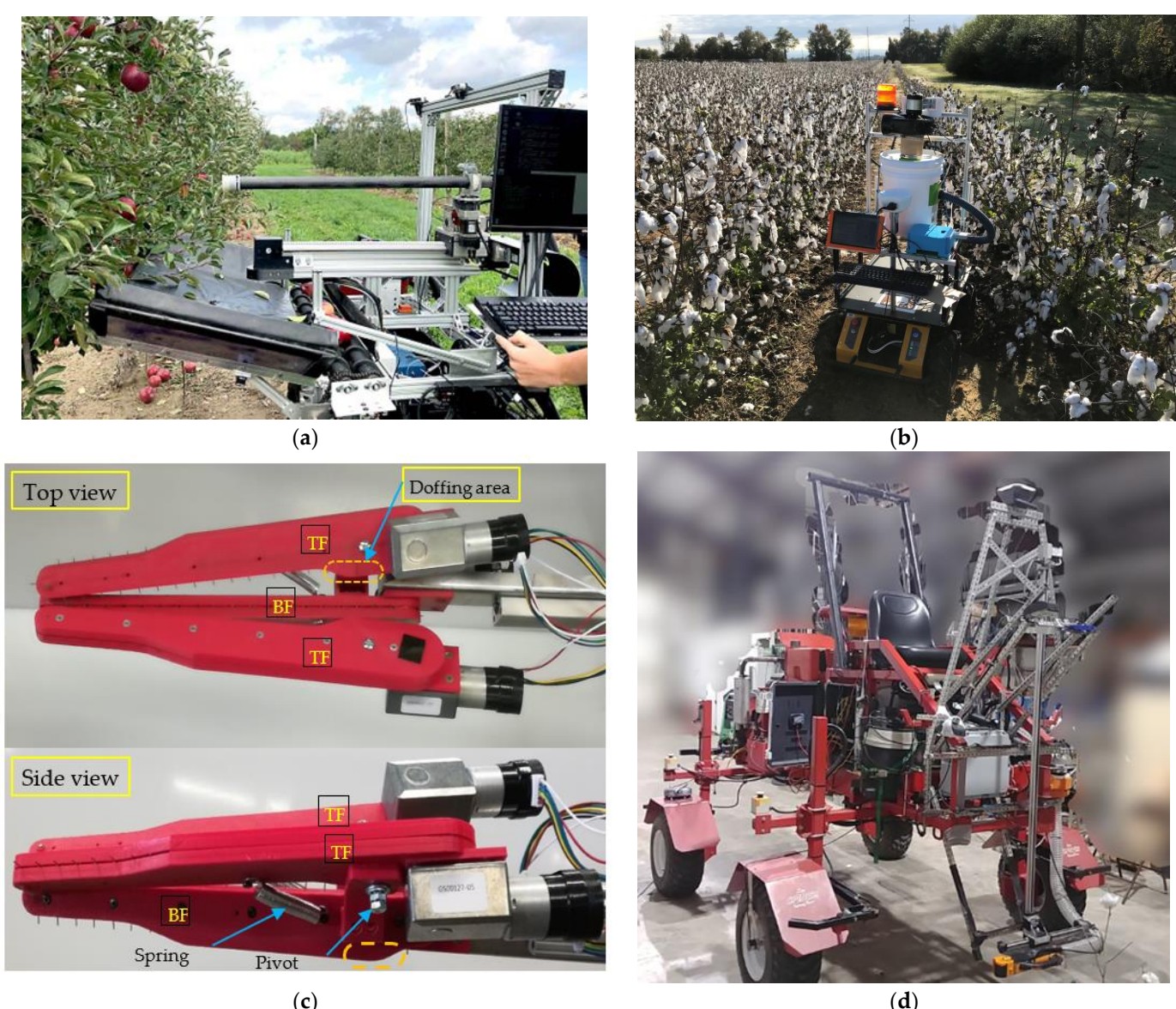

**Figure 3.** Some of the agricultural robots with vacuum suction and plucking concepts: (**a**) robotic apple harvesting prototype [33]; (**b**) cotton harvesting autonomous platform [36]; (**c**) the three-finger configuration of the end effector for harvesting cotton; TF: top finger; BF: bottom finger [38]; (**d**) center-articulated hydrostatic cotton harvesting rover [39].

Fue et al. [39] installed a 2.5 hp wet/dry vacuum cleaner on the rover (Figure 3d). The vacuum cleaner was connected to the end ejector using a 90 cm flexible plastic line to assist in picking and moving the picked cotton bolls. The cotton bolls were sucked into the hose, which was placed near the cotton bolls. The hose was placed near the cotton bolls, and the cotton balls were sucked into the hose. A rotating brush roller at the final discharge used vibration, rotation, and sweeping motions to capture and extract the cotton bolls. The motor driving the brush roller had a voltage of 12 VDC. The cotton bolls were conveyed by a flexible hose to a porous impeller equipped with the suction inlet of the vacuum. The cotton bolls were dropped into a bag as the porous impeller [39]. Table 2 summarizes the details of vacuum suction and picking in apple and cotton harvesting.

**Table 2.** Summary of the information about vacuum suction and plucking in apple and cotton harvesting.

| Crop | End Effector Types | Mechanism | Advantages | Disadvantages | Cited Work |
|---|---|---|---|---|---|
| Apples | End effector with a soft silicone vacuum cup and rotating mechanism | Vacuum suction | Effective and adjustable to various apple forms; minimizes fruit damage; efficient detaching maneuver | None reported | [33] |
| Cotton | One DOF cartesian manipulator with a vacuum suction end effector | Vacuum suction | The mobile platform can move around the field and pick cotton quicker | Unsatisfactory performance due to varying suction distance of boll/locks to harvester nozzle | [36] |
| Cotton | Mobile platform with a rolling stripper and a suction motor | Rolling stripper and suction | None reported | Research ongoing to determine effectiveness as once-over harvester | [37] |
| Cotton | Three-finger moving pinned belt end effector | Moving pinned belt | Complies with all design criteria; clean and complete picking | Consumes calyx of the boll during picking failures; longer picking time | [38] |
| Cotton | End effector with a spinning brush roller and a 2.5 hp wet/dry vacuum | Vacuum suction and sweeping | Effectively picks up and moves cotton bolls | None reported | [39] |

### 2.3. Twisting and Pulling

Twisting and pulling for apple harvesting features the end effector comprises three pneumatic actuators that are highly compliant and positioned around a flexible, soft palm that supports an apple against the actuators. The end effector is equipped with three actuators, the upper two of which measure 95.25 mm and serve to stabilize the apple during grasping. The lower actuator is longer and measures 152.4 mm. It is designed to enclose the apple to prevent interference with smaller apples (Figure 4). The actuators were printed with a larger cross-sectional area and no alignment constraints, resulting in a higher traction force at a given input pressure. However, the robot must be repositioned, or it only will harvest the identified apples, leaving other hidden fruit on the tree [40].

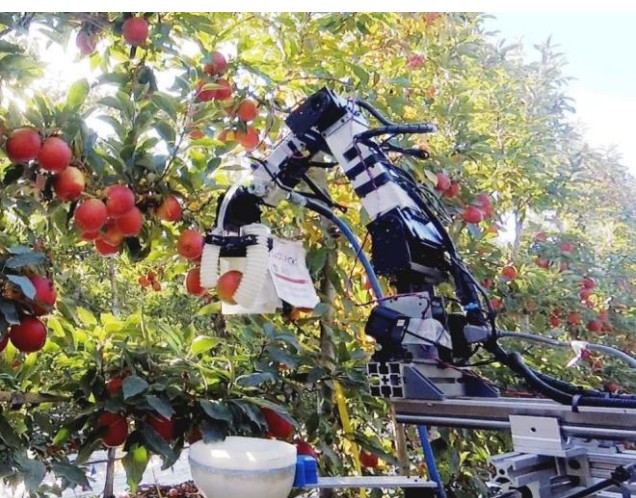

**Figure 4.** Robotic apple harvesting system with a 3D-printed soft robotic end effector [40].

De Preter et al. [41] designed a robotic arm with fingers that have a soft 3D-printed frame structure for gripping strawberries. This structure creates a larger contact area that results in better pressure distribution compared to human fingers. The gripper gently pulls the fruit out of the plant with circular motions. The robot arm and the gripper perform this movement in a similar way.

In mango harvesting, the robot's gripper holds the fruit and performs either a twisting or pulling motion, or a combination of both, to release it from the branch. This method produces similar results to hand picking. When a mango is harvested, the small stem that connects it to the tree often falls off, and the fruit secretes a sap-like substance that covers the exposed area, possibly accelerating its decomposition. These observations led Khare et al. [42] to develop a concept for a mobile robot equipped with a manipulator to grasp the fruit.

For kiwi, Williams et al. used tubes/chutes to transport the harvested fruit after separation. This decreased the cycle time since the arms were no longer required to set fruit down after harvesting. Kiwifruit can be pulled away from the canopy by simply clasping it and tugging; this exerts too much power on the fruit and shakes the canopy unnecessarily. Other fruit swing when shaken, accidentally separating them from the surrounding fruit. Movement in the canopy impacts the accuracy of previously calculated fruit locations produced by the vision system. In addition, it raises the probability of ripping the stem from the plant rather than the fruit, raising the risk of damage during storage and transportation. The robot rotates the fruit upwardly around the stem before tugging it downwards to lessen the power needed and encourage stem removal. As a result, the force is concentrated at the fruit–stem junction, which promotes ripping there. This method of fruit removal is comparable to what commercial kiwifruit pickers do. The end effector shown here uses an asymmetrical four-bar connection to achieve the rotation. A 3D-printed set of digits are molded into food-grade silicon to create the clasp mechanism. When held, the channeled air pockets in the molded silicon parts enable the silicon to take on the shape of the kiwifruit [43].

Another study describes the development and evaluation of an end effector specifically designed for harvesting kiwifruit grown using scaffold techniques. The end effector utilized these discoveries, which implemented integrated pickup and unloading of kiwifruit using bionic fingers and a converse cam mechanism [44]. Table 3 summarizes the details of the twisting and plucking in apple, strawberry, mango, and kiwi harvesting.

**Table 3.** Summary comparison table for the twisting and pulling in apple, strawberry, mango, and kiwi harvesting.

| Crop | End Effector Type | Mechanism | Special Features | Cited Work |
|------|-------------------|-----------|------------------|------------|
| Apple | Three pneumatic actuators | Twisting and pulling | Highly compliant, flexible, soft palm, reduced actuator interference, higher pulling force, requires tree reimaging | [40] |
| Strawberry | Soft 3D-printed framework structure | Twisting and pulling | Larger contact surface, gentle pulling with circular motion compared to human fingers | [41] |
| Mango | Gripper | Twisting and pulling | Twists and/or pulls fruit, inspired by preventing sap exposure and decomposition | [42] |
| Kiwi | Asymmetrical four-bar connection | Rotation and pulling | Food-grade silicon bionic fingers, air pockets to take on fruit shape, integrated pickup and unloading using converse cam mechanism, reduces canopy movement, concentrates force at fruit–stem junction | [43] |

*2.4. Shaking and Collecting*

An apple harvesting technique involving shaking and collection was also investigated. A control console allows manual adjustment of travel speed and direction. The shaker used in this technique was modeled after a commercial hand shaker (SP200, STIHL Inc., VA, USA), which has a V-shaped hook head and a linear travel stroke of 3.6 cm. It was mounted on a sliding system so that it could reach the desired limbs. Two three-level support frames (catch boards) and six pieces of buffer foam were used to create a fruit-catching surface (Figure 5). The dimensions of each catching surface were 2.5 m × 1.2 m (length × width) and had an adjustable angle of inclination [45].

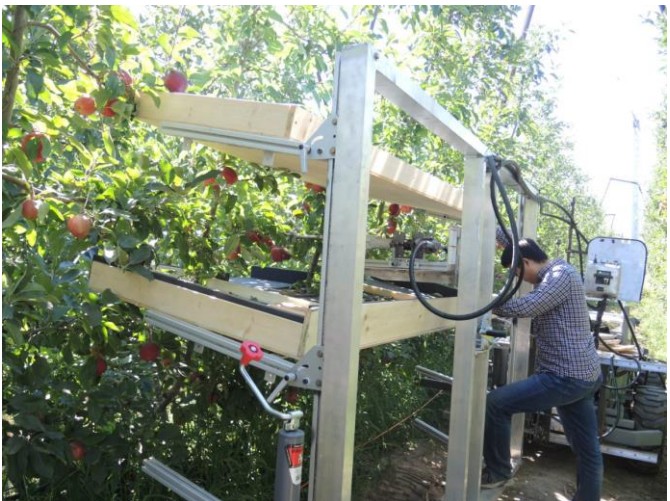

**Figure 5.** A multitier shake-and-catch apple harvesting platform [45].

In their study, He et al. [45] investigated a multistage shaking and trapping system with two shaking sites. The first site was shaken at the center of a branch and included two adjacent branches at the same level from two adjacent trees. The second site was shaken at the junction of the branch and trunk and included two branches at the same level of the same tree. Each branch was shaken twice on the first site, while only once on one of the two branches near the trunk on the second site. The researchers also examined the effect of shaking duration on harvester performance using two different durations of 2 and 5 s. The results showed that 2 s shaking was sufficient to remove most fruit in the study but shaking in the second site caused significantly more fruit damage.

When comparing the two shaking methods, it is important to note that there are many different factors that can affect the results. For example, the mechanical and platform differences between the linear and nonlinear shaking techniques can have a significant impact on the results. One possible explanation is that the abrupt interruptions in fruit movement caused by intermittent shaking can cause the fruit to tilt and spin, which can increase the separation force on the abscission layer [45].

Fruit removal efficiency peaked within a short period of continuous shaking, and increasing the duration of shaking would not improve results. However, intermittent shaking resulted in slightly reduced fruit removal efficiency compared to continuous shaking. This was due to the sudden interruptions in the movement of the fruit. In addition, fruit quality was evaluated, and it was found that intermittent linear shaking resulted in a slightly lower percentage of marketable fruit (85.3%) than continuous linear shaking (88.2%). Although the difference was small, longer shaking duration increases the probability of fruit–branch collisions [45].

In their study, Zhang et al. [34] provided valuable insights into the development of shaking and catching systems for apple harvesting using a field evaluation of targeted shaking and catching. However, a limitation of the study is that a direct comparison of continuous nonlinear and intermittent linear shaking methods is difficult because the data

were collected from two different apple cultivars ('Gala' and 'Scifresh') and it is known that cultivars can affect harvest results. The study recommended a longer shaking time for the shaking and catching method as it did not significantly affect fruit quality and only slightly improved fruit removal efficiency; a shaking time of 5 s is recommended.

Shaking and catching involves loosening apples by shaking the trunk or branches of an apple tree and then using special equipment to catch them as they fall to the ground. However, they could not prevent bruising brought on by collisions between apples, trees, and containers; therefore, the apple industry did not implement them [34].

Growers in Washington frequently use apple trees with SNAP (simple, narrow, accessible, and productive) structures. Because of their small and thin canopies, vertical SNAP structures were chosen for this study because they offered options for shake and catch harvesting with optimal shaking and localized capturing [34].

Researchers have tried to employ shake-and-catch systems, which vibrate the entire plant, as we described in the opening of this article. Results from this method of bulk fruit picking were unacceptably poor for fruits intended for the fresh market. Future research may focus on shaking small, specific parts of the tree in orchards of trellis apples or other fruits in order to detach and harvest the fruit [14].

De Kleine et al. [46] have developed a new tool for applying different shaking patterns and rhythms to fruit set. This tool, called the dual motor actuator (DMA), was developed based on an idea from Washington State University and uses two eccentrics connected by two arms to manipulate the trajectory applied to a point. This mechanism allows the creation of an infinite number of rhythmic patterns within a given physical space. The DMA can generate patterns at a frequency of 10 to 300 cycles per minute (cpm) and consists of two electric motors, two stepper motor drivers, and two power supplies. To create rhythmic patterns, the speed and direction of each motor are adjusted, which is controlled by an Arduino Mega using pulse-width modulation. The use of stepper motors allows for easy programming to create different shaking patterns. Table 4 summarizes the details of the shaking and collecting technique for apple harvesting.

**Table 4.** Summary comparison table of the shaking and collecting technique for apple harvesting.

| Crop | End Effector Type | Mechanism | Special Features | Cited Work |
|---|---|---|---|---|
| Apples | Handheld shaker adapted with V-shaped hooking head and linear motion stroke | Shaking and collecting technique with a sliding system, two three-tier supporting frames, and six pieces of buffering foam | Two shaking locations were tested, Section 1 (middle of a limb) and Section 2 (branch-trunk junction), two different shaking durations (2 and 5 s), intermittent shaking resulted in a slightly decreased fruit-capturing efficiency and lower proportion of marketable fruit | [45] |
| Apples | Dual motor actuator | Grasping and shaking with various combinations of shaking patterns and rhythms | Can apply patterns to limbs at frequencies ranging from 10 to 300 cpm | [46] |

## 3. System Components

Robotic systems are characterized as programmable mechanical devices that interact with their environment, including people, using numerous sensors, actuators, and human interfaces to perform a specific task. Agricultural robots are typically built to perform a variety of tasks, such as planting, weeding, pruning, picking, harvesting, packing, and handling. To perform the required task, the robot must be equipped with mechanical, sensing and control systems, including mobility, steering and control, manipulators, end effectors, path planning, navigation, and target detection and recognition systems [47]. The components of the agricultural robot system were divided into four subsystems: mobile platform, manipulators and end effectors, sensors and localization, and path planning and navigation (see Figure 6).

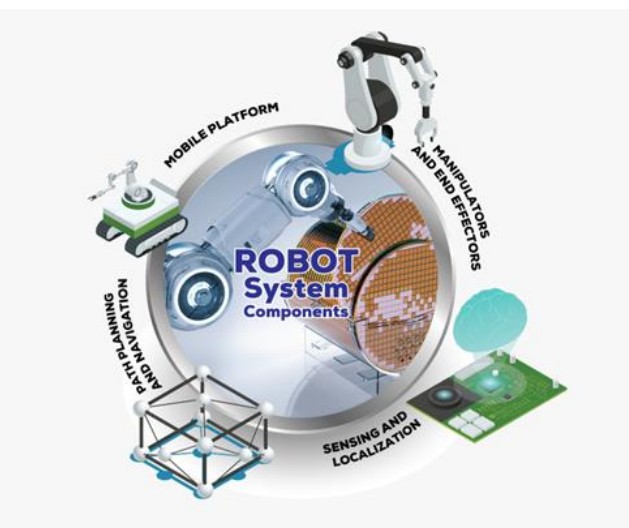

**Figure 6.** Four subsystems in agricultural robot systems.

*3.1. Mobile Platform*

Originally, unmanned ground robots were produced in agriculture by retrofitting existing commercial tractors, but there is now a trend toward creating mobile platforms specifically designed for robotic purposes. These platforms can be divided into two types: four-wheel platforms with two- or four-wheel drive and two- or four-wheel steering, and platforms with tracked or six-wheel drive. Several factors must be considered when designing an agricultural robot, including the ability to operate in wet conditions without becoming stuck or damaging the ground, maintaining an economically feasible cost structure, and using a flexible robot frame or platform that keeps all wheels in contact with the ground while minimizing complexity [48].

Several researchers [34,44,45] examined a self-propelled orchard platform with a four-cylinder diesel engine and hydraulic four-wheel drive as the base platform for shake-and-catch apple harvesting research. To allow for ground operation, the control panel was relocated, and the worker platforms were removed. The driver could manually adjust the vehicle's speed and direction using an operational console. Several researchers presented using an articulated steer tractor for developing the cotton harvesting robot [49–51]. Another study using an autonomous tractor as a mobile platform was conducted for harvesting heavyweight crops such as pumpkins and watermelons [32,52].

The majority of the reported harvesting robots are four-wheel vehicles. The development of specifically designed mobile platforms has been carried out by using a commercial platform for easy mobility in field environments. Zhang et al. [33] developed a robotic prototype for apple harvesting based on a Segway mobility platform (Figure 7a) and equipped with three modules: an Intel RealSense camera, a 3 DOF manipulator, and a vacuum-based end effector. The Husky A200 from Clearpath Robotics (Figure 7b) was the mobile platform for the cotton harvesting project [36]. The platform's width of 68 cm fits standard cotton row spacings, making it appropriate for field operations.

In contrast to large farm machinery, it is lightweight for field traffic and does not cause soil compaction [53]. The platform can move at speeds of 1 m per second and is strong enough to support weights of up to 75 kg. It features a 24 V DC lead–acid battery that can run the device for two hours. Up to 3 h of operation are provided by two new lithium polymer batteries, each having six cells and a 10 Ah rating [36].

A specially created platform (Superdroid, Inc., NC, USA) was utilized for sugar snap pea harvesting as a movable basis for the robot [54]. The two 12 V-18 Ah sealed lead–acid batteries, and four 24 V DC motors added up to about 30 kg for the aluminum mobile platform. It could carry a cargo of up to 30 kg and move at a maximum forward speed of 6 km/h (with the selected transmission). All four motors, two left and two right, were

linked in pairs to drive the movable base. Differential steering or changing the speed of the left and right motor pairs was used to turn the vehicle. In addition to the height that the mobile robot can occupy, a low profile was developed for the kiwi harvesting robot so that the modules held by the platform have a clearance between them and the canopy at a height of 1.4 m to 1.7 m [42,55].

Tracks platforms permit operation in various environmental situations, including muddy fields and wet soil following major precipitation occurrences. Compared to wheeled robots, broad tracks reduce the physical effect on the soil. The TERRA-MEPP robotic platform (Figure 7c) was developed as a comprehensive high-throughput plant phenotyping solution for the production of biofuels from energy sorghum [56]. The robotic platform TERRA-MEPP can move along a single row with some degree of autonomy and capture the crop from multiple angles above, below, and adjacent to the canopy. Another tracks platform was reported for the apple harvesting robot application [57].

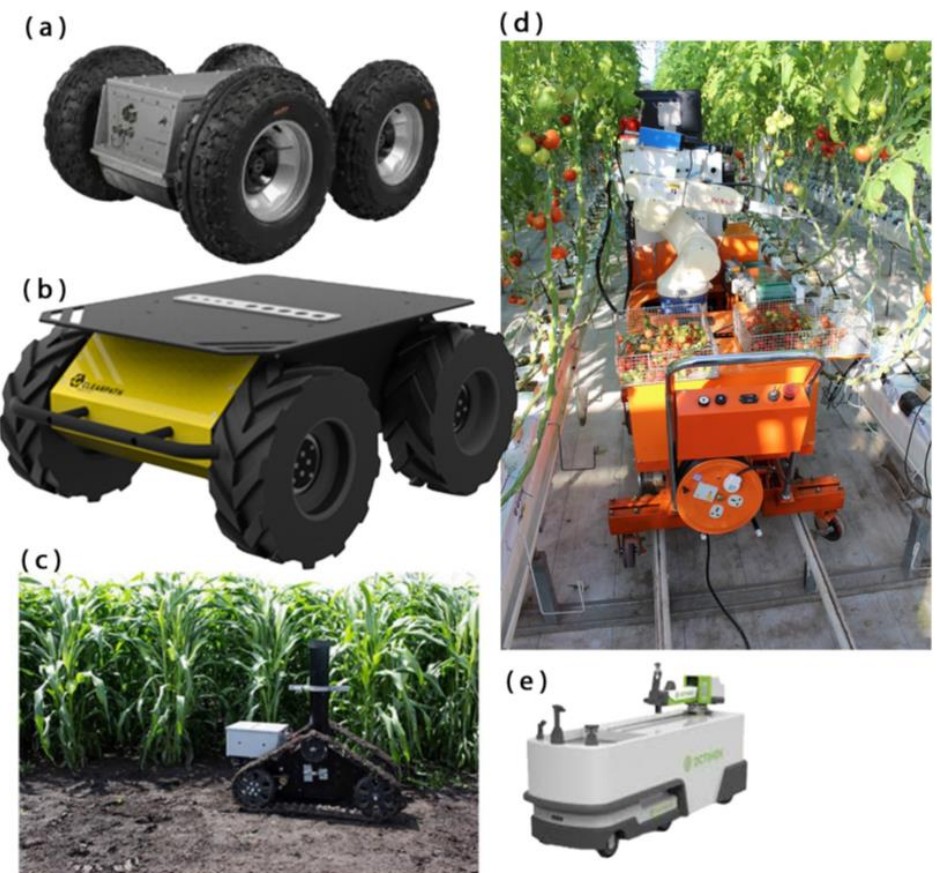

**Figure 7.** Some of the agricultural robots' mobile platform types: (**a**) railed platform [22]; (**b**) tracked platform (TERRA_MEPP) [56]; (**c**) independent steering devices (Octinion) [41]; (**d**) four-wheel platform (Seqway) [33]; (**e**) four-wheel platform (Husky A200) [36].

Several studies have been presented using the railed vehicle robot platforms for harvesting purposes in the greenhouse. Tomato harvesting [58,59], cherry tomato harvesting (Figure 7d) [22], strawberry harvesting [60,61], and sweet paper harvesting [26] have all been explored using the railed vehicle robot platforms in a guided rail. Other researchers developed a mobile platform using four wheels in the greenhouse. Lehnert et al. [27] presented Harvey, a novel harvesting robot for the autonomous harvesting of peppers in protected growing environments. Wang et al. [62] designed a tomato harvesting robot with an independent four-wheel steering system. De Preter et al. [41] developed Octinion (Figure 7e), the first robot that can pick strawberries similar to a human and specifically designed for tabletop systems.

These recent innovations offer further advantages in agility, crop and terrain adaptation, and flexibility. Although wheeled or legged designs offer comparable advantages and adaptability to different crops and terrains, mobile robots with independent steering devices offer the best mobility on uneven or sloping terrain. Intelligent farms will require ground mobile robots, which should be created using robotics concepts and drawing on the experience acquired in creating traditional tractors [9].

Although several working semicommercial systems were created and the mobility and steering capabilities thoroughly explored, there needs to be more information available comparing different types of systems or their suitability for agricultural land. Table 5 summarizes the overview of mobile platforms for agricultural robots.

**Table 5.** Overview of mobile platforms for agricultural robots.

| Mobile Platform Type | Characteristics | Applications |
| --- | --- | --- |
| Four-wheel platform with two or four-wheel drive and two or four-wheel steering | Lightweight, flexible frame, and suitable for wet conditions without harming soil structure. | Cotton harvesting, pumpkin and watermelon harvesting, apple harvesting |
| Tracked platform or six-wheel drives | Reduces physical effect on soil, suitable for various environmental situations. | Energy sorghum phenotyping, apple harvesting |
| Railed vehicle robot platform | Guided rail system for greenhouse harvesting. | Tomato harvesting, cherry tomato harvesting, strawberry harvesting, sweet pepper harvesting |
| Independent steering devices | Finest mobility in sloped or irregular terrain. | Strawberry picking, tomato harvesting, sugar snap pea harvesting, kiwi harvesting |

### 3.2. Manipulators and End Effectors

In agriculture, an arm manipulator is commonly used to position its end effector accurately and with the required orientation to interact with an object. Depending on the specific task, the manipulator can be equipped with a suitable end effector. Several types of manipulators provide different characteristics, such as rigid, soft, parallel, dual-arm, redundant, hyper redundant, or continuum manipulators [9].

This literature has reported different degrees of freedom (DOF) for manipulators, from a three-DOF rectangle coordinate manipulator to a seven-DOF manipulator. For three-DOF manipulators, several autonomous robots have been developed for harvesting strawberries [41], sweet pepper [63], kiwifruit [42,55], and cotton [38]. Zhao et al. [58] designed and tested a two-armed frame with two three-DOF manipulators and two types of end effectors (a saw-like knife and a suction device) for tomato harvesting. Ling et al. [59] also used a dual arm with a suction cup and cutting gripper. A single-rail dual-arm manipulator with two grippers was reported to work more accurately and faster than the previous version with a five-DOF manipulator [61].

Another research presented on harvesting pumpkin [32], strawberry [23,61], apple [40], and sugar snap pea [54] by using a five-DOF. Most of the researchers that we found in this literature developed harvesting robots by using a six-DOF manipulator for cherry tomato [22], tomato [62,64], apple [14,65], sweet pepper [27], citrus [31], eggplant [29], and iceberg lettuce [28]. A grape harvesting manipulator has been developed with a seven-DOF to which two different end effectors can be attached for harvesting and green harvesting, and another for defoliation [19]. Table 6 summarizes the overview of manipulators and end effectors for agricultural robots.

**Table 6.** Overview of manipulators and end effectors for agricultural robots.

| Number of DOF | Type of Manipulator | Applications |
|:---:|:---:|:---:|
| 3 | Autonomous robots with single-arm manipulators | Harvesting strawberries, sweet pepper, kiwifruit, and cotton |
| 3 | Dual-arm frame with two three-DOF manipulators | Picking tomatoes with saw-type cutter and suction device |
| 3 | Dual-arm with a suction cup and cutting gripper | Harvesting tomatoes |
| 5 | Manipulator with five-DOF | Harvesting pumpkin, strawberry, apple, and sugar snap pea |
| 6 | Manipulator with six-DOF | Harvesting cherry tomato, tomato, apple, sweet pepper, citrus, eggplant, and iceberg lettuce |
| 7 | Manipulator with seven-DOF | Grape harvesting with end effectors for harvest and green harvest, and defoliation |

### 3.3. Sensing and Localization

One of the essential components of mobile robotics is the sensor. Sensors provide mobile robots the ability to carry out tasks, including trajectory tracking, target location, and tracking, acting safely by avoiding collisions, and localizing and mapping the surroundings, among other tasks [7]. To increase actuation, world modeling, and reasoning integrate, modify, and augment sensor output [66]. Although they are essential, they also provide a constraint for mobile robots since it is sometimes impossible to find excellent, reliable, and accessible sensors that can accurately determine a robot's location. Most agricultural robots currently use global navigation satellite systems (GNSS), GPS, and image sensing systems to localize themselves [19,32,52,55,56,61].

Sensors commonly used in agricultural robotics are tactile, force torque, encoders, infrared, ultrasonic, sonar, active beacons, accelerometers, gyroscopes, laser range finders, vision-based, color tracking, contact and proximity, pressure, and depth sensors [67]. Researchers typically use stereo cameras with multiple lenses and a separate image sensor or film image for each lens to determine plant locations [22,27,33,42,58,61,64,65,68]. A real-time object identification system called YOLOv3 (You Only Look Once, Version 3) recognizes things in films, live feeds, or still photos. To find an item, the YOLO machine learning system leverages characteristics that a deep convolutional neural network has learned [38,64,69].

A robot operating system (ROS) is widely used in a compact robot to communicate and transmit data between software and hardware components. ROS is a global open-source framework that has helped robotics to expand recently in agriculture applications. Most researchers use either Phyton or C++ as the programmable language [27,33,52]. In ROSs, there are five primary modules: the visual perception system, the state estimation, which consists of localization and mapping, the obstacle detection, the task execution, and the navigation [19]. The deployment of agricultural mobile robots is anticipated to increase due to the availability of open-source platforms and the decline in sensor costs. Table 7 summarizes the overview of sensing and localization for agricultural robots. Figure 8 shows some basic sensing and localization components typically used in agricultural robotic systems.

**Table 7.** Overview of sensing and localization for agricultural robots.

| Component | Description |
|---|---|
| Sensor | It provides robots with the ability to perform tasks and perceive the environment. Tactile, force torque, encoders, infrared, ultrasonic, sonar, active beacons, accelerometers, gyroscopes, laser range finders, vision-based, color tracking, contact and proximity, pressure, and depth sensors are commonly used in agricultural robotics. |
| Localization | Determines a robot's location in the environment. Global navigation satellite systems (GNSS), GPS, and image sensing systems are widely used in agricultural robots. |
| Object identification | YOLOv3 is a real-time object identification system used to recognize specific things in films, live feeds, or still photos. |
| Programming language | Python and C++ are commonly used as the programmable language in agricultural robotics. |
| ROS modules | ROS has five primary modules: visual perception system, state estimation (localization and mapping), obstacle detection, task execution, and navigation. |
| Open-source platforms | The availability of open-source platforms and the decline in sensor costs are anticipated to increase agricultural mobile robot deployment. |

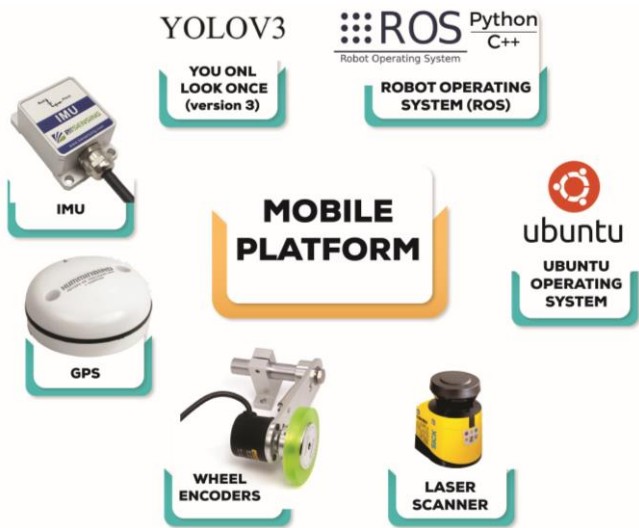

**Figure 8.** Some basic sensing and localization components in agricultural robot systems.

### 3.4. Path Planning and Navigation

Finding a continuous route for the robot to travel from the initial state to the target state/configuration is called path planning [50]. To perform path planning, the mobile system uses a known map of the environment stored in the robot's memory. The state/configuration provides the robot with a possible position in the environment, and it can move from one state/configuration to another by performing various actions [7]. Path planning is a crucial aspect of robot control and must be collision-free, reachable, smooth, safe, predictable, and responsive to enable robot integration in industry and society [9]. Path planning algorithms are used by autonomous vehicles, unmanned aerial vehicles, and mobile robots to determine safe, effective, collision-free, and cost-efficient paths from the starting point to the destination [70].

There may be one or more paths or no path at all between the specific start and destination states/configurations that connect the states/configurations. Additional requirements or criteria are often introduced to specify the preferred optimality, e.g., shortest length,

shortest travel time, least distance to obstacles, smoothness without abrupt turns, and minimum movement constraints [71]. In fruit harvesting, path planning is determined by manipulators, end effectors, and the type of agricultural produce being harvested. When using an arm with many degrees of freedom (DOF), path planning is computationally intensive. However, most path planning methods are more effective and successful when the number of DOF is kept small enough to achieve the goal [68].

Robot navigation refers to the ability of a robot to determine its location in its environment and plan a route to a specific destination. In order to navigate, a robot needs a map of its environment and the ability to interpret it. Path tracking algorithms are commonly used by robots in open fields using GPS and cameras, while robots in greenhouses are often guided by tracks and therefore require position control algorithms instead of navigation algorithms [19,32,52,56,58–61]. Some studies have explored motion planning using arm movements without including search mechanisms for obstacle avoidance or path planning [42,55,57]. A recent study has introduced path planning in agriculture by using advanced methods in convolution neural networks (CNN) [33,65], as well as navigating on predetermined paths in maps [19,52]. Table 8 summarizes the overview of path planning and navigation in agricultural robotics.

**Table 8.** Overview of path planning and navigation in agricultural robotics.

| Topic | Summary |
|---|---|
| Path Planning | Finding a continuous path for the robot from start to goal state/configuration. It remains one of the fundamental aspects of robot control. Path planning algorithms are used by mobile robots, UAVs, and autonomous cars. Additional criteria are introduced to define desired optimality. Path planning in fruit harvesting depends on manipulators, end effectors, and produce. |
| Robot Navigation | Robot's ability to determine its position and plan a path towards the goal location. It requires a map of the environment and the ability to interpret it. Most robots apply path-tracking algorithms using GPS and cameras, while greenhouse robots use rails. Motion planning using arm trajectory is also used in some cases. Advanced methods like convolution neural networks are also being used for path planning in agriculture. |

## 4. Discussions

Given the challenges and opportunities discussed in the previous sections, the development of a cotton harvesting robot requires a comprehensive understanding of the various subsystems that enable its operation. The cotton plant is characterized by being grown as a fiber crop for its soft, fluffy, and fibrous bolls. Cotton plants are generally grown in warm, humid climates and require a lot of water for growth. The cotton bolls are harvested and processed into cotton fiber, which is used to make a variety of textile products such as clothing, bedding, towels, and more. The design concept for such a robot requires careful consideration of the mobile platform, manipulators and end effectors, sensing and localization, and path planning and navigation.

### 4.1. Mobile Platform

The mobile platform plays a critical role in the efficiency and effectiveness of the cotton harvesting robot. It is important to choose a platform that is capable of negotiating rough terrain and uneven surfaces, which are common in cotton fields. In addition, the platform must be able to support the required weight of the robotic components, including manipulators and end effectors. The development of mobile platforms for unmanned ground vehicles (UGVs) in agriculture is a critical area of research that involves the development of specially designed platforms based on robotics principles. These platforms must address several factors, including the need to operate in wet conditions, keep costs low, and reduce complexity. Most harvesting robots are four-wheeled vehicles, but the use of tracked platforms allows them to operate in a variety of environmental situations,

including muddy fields and wet soils. In addition, robotic platforms with tracked vehicles have been explored for greenhouse harvesting purposes. Recent innovations in mobile platforms offer further advantages in agility, crop and terrain adaptation, and flexibility. Mobile robots based on independent steering devices have the best maneuverability on sloping and irregular terrain. However, information on comparisons between the different types of systems or their applicability to agricultural terrain is rarely available. Overall, the development of mobile ground robots for agriculture is critical for smart agriculture and should be done using robotic concepts and taking advantage of the experience gained in the development of conventional tractors.

### 4.2. Manipulators and End Effectors

Manipulators and end effectors are critical components that allow the robot to interact with and harvest cotton plants. Manipulators must be designed to reach cotton plants and adapt to different plant heights and positions. End effectors must be designed to grip and pull off cotton bolls without damaging the plant or cotton. Arm manipulators that are able to move their end effector to a specific location with the required orientation to interact with an object are increasingly being used in agriculture. The type of end effector used depends on the agricultural task at hand. Different types of manipulators may have unique characteristics, such as rigid, soft, parallel, dual-arm, redundant, hyper redundant, or continuous manipulators. The literature describes different degrees of freedom for manipulators ranging from a three-DOF manipulator with rectangular coordinates to a seven-DOF manipulator. The literature provides an overview of different types of manipulators and end effectors for agricultural robots with different degrees of freedom. It is evident that different agricultural tasks require different types of manipulators and end effectors to be most effective.

### 4.3. Sensors and Localization

Sensors and localization are critical components for the cotton harvesting robot to navigate and avoid obstacles in the field. The use of sensors is an essential aspect of mobile robotics, allowing robots to perform various tasks such as tracking, navigation, and collision avoidance. Agricultural robots also rely heavily on sensors, as they require precise localization and mapping to operate effectively in an agricultural environment. However, it can be difficult to find reliable and easily accessible sensors. Therefore, researchers have used a variety of sensors, including tactile, force, torque, encoder, infrared, ultrasonic, sonar, active beacons, accelerometers, gyroscopes, laser rangefinders, vision systems, color tracking, contact and proximity sensors, and pressure and depth sensors. In particular, image sensor systems, including stereo cameras and machine learning-based object recognition systems, are widely used to estimate the location of plants and identify specific objects.

### 4.4. Robot Operating System

The robot operating system (ROS) has also become a popular framework for robot development, with many researchers using either Python or C++ as the programming language. ROS offers five main modules, including visual perception, state estimation (localization and mapping), obstacle detection, task execution, and navigation. With the use of open-source platforms and decreasing sensor costs, the use of mobile agricultural robots is expected to increase in the coming years. The integration of sensors and localization technologies is critical to the development of effective agricultural robots. Researchers have used a wide range of sensors and machine learning-based systems to achieve accurate crop localization and identification, while ROS provides a robust framework for robotic development. It is expected that these advances in sensor technology and robotics will lead to more efficient and cost-effective agricultural practices.

### 4.5. Path Planning and Navigation

Path planning and navigation are critical components to ensure that the cotton harvesting robot can navigate fields efficiently and safely. Path planning algorithms must be designed to determine safe and efficient travel paths while avoiding obstacles and minimizing damage to crops. Navigation algorithms must allow the robot to accurately determine its position and plan a path to the cotton plants. Robot navigation requires the robot to determine its position in its reference frame and then plan a path to its destination. Navigation systems rely on representations of the environment, such as maps, and the ability to interpret those representations. Various sensors, such as GPS, cameras, and rails, can be used to track the robot's position and navigate in the environment. In addition, there are recent studies that have introduced advanced methods, such as convolutional neural networks (CNN), for path planning and navigation in agriculture. The complexity of path planning increases when the robot has many degrees of freedom (DOF) to fulfill its purpose, such as in fruit harvesting. However, most path planning methods are more effective and successful when the number of degrees of freedom is optimized to be small enough. Therefore, the selection of appropriate path planning and navigation strategies for mobile robots is crucial to ensure safe and efficient integration in the agricultural industry.

### 4.6. Limitations and Gaps in the Field of Harvesting Robots

The development of harvesting robots has come a long way, but there are still many obstacles and gaps in technology. Some of the major limitations and gaps are as follows:

1.  Crop variability: Harvesting robots need to be able to recognize and select different types of crops in different situations, such as different maturity levels and sizes. However, given the diversity of crops, this can be a difficult task, especially when dealing with complicated and delicate fruits and vegetables.
2.  Complicated environments: Farms often have complex environments with a wide range of topography, lighting, and weather. These difficult environments require harvesting robots to function, which can be a significant barrier to their growth.
3.  Cost: Harvest robots are currently quite expensive to develop and produce, which could discourage many farmers from using them. This could discourage smaller farmers from using harvesting robots.
4.  Restricted crop varieties: Currently, harvesting robots are best suited for uniform and easy-to-harvest crops such as strawberries and lettuce. However, they may not be suitable for crops that require more specific harvesting methods.
5.  Safety issues: The development of harvesting robots must consider safety for both the crops being harvested and the workers who are near them. Their success depends on being able to work safely in the agricultural environment.
6.  Social acceptance: some farmers and customers may be opposed to the use of harvesting robots because they are concerned about how it will affect their jobs and conventional farming practices.

Addressing these limitations and shortcomings is critical to the continued development and use of harvesting robots in agriculture. Some of these limitations will likely be addressed as technology evolves, but this will take time and continued investment in research and development.

### 4.7. Summary

In summary, the development of a cotton harvesting robot requires careful consideration of the mobile platform, manipulators and end effectors, sensing and localization, and path planning and navigation. Each of these components plays a critical role in the efficiency and effectiveness of the robot, and their design must be optimized to work together seamlessly to ensure a successful cotton harvest.

Therefore, our proposed proof of concept design of an autonomous cotton harvesting robot consists of a single-row harvesting module with a set of finger roller cotton stripper mechanisms attached to the front of the Amiga platform from Farm-NG. The Amiga

platform is an all-electric 4 × 4 skid steer that can be configured for a differential drive. The height and width of the platform can be easily adjusted to suit different crops. The design includes a mechanism for pulling the cotton plant and then transporting the harvested bolls to the bucket in the center of the platform using suction motors and hoses. The selective harvesting function is enabled by use of two electric cylinders that control the retraction and extension movements of the harvesting modules (Figure 9).

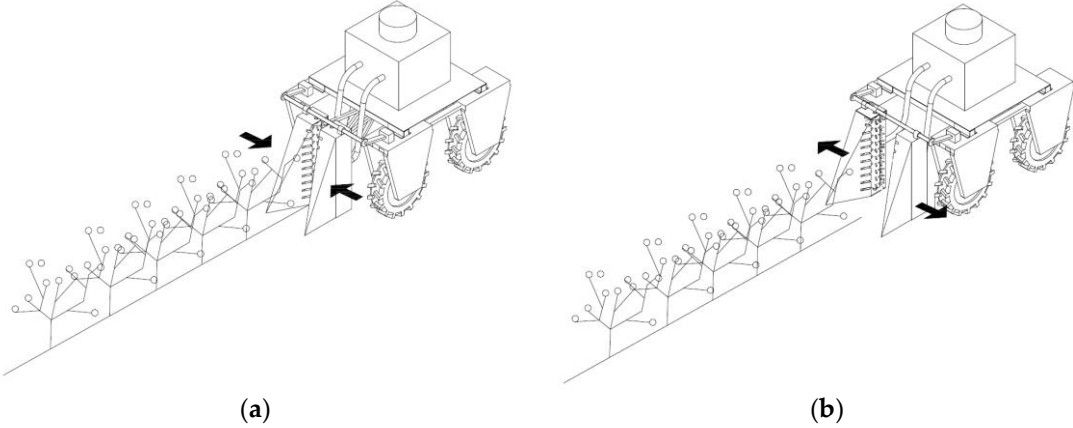

(**a**)   (**b**)

**Figure 9.** (**a**) Extension movements of the harvesting modules; (**b**) retraction movements of the harvesting modules.

Overall, the proposed design is a promising approach for the development of an autonomous cotton harvesting robot, as it has a number of unique features that could significantly improve the efficiency of the harvesting process. The use of a single-row harvesting module in combination with the finger roller cotton stripper mechanism enables a precise and targeted approach to cotton plant harvesting. The transport mechanism with suction motors and hoses helps to ensure that the harvested bolls are not damaged during the harvesting process.

Moreover, the selective harvesting function enabled by the electric cylinders provides an additional advantage, as the robot can selectively harvest only mature bolls, improving the overall yield and quality of the harvested cotton. However, it should be noted that the proposed design has not yet been tested in the field. Further research and testing are needed to evaluate the effectiveness and feasibility of this concept under real harvesting conditions. In addition, potential challenges such as variations in cotton plant size and growth patterns, as well as possible technical problems with the electric cylinders or other components of the harvesting module, must be considered.

## 5. Conclusions

In conclusion, harvesting machines have become established for crops such as wheat, soybeans, and corn, while fruit and horticultural crops are still mostly harvested manually. However, recent research has investigated various mechanisms for harvesting fruit, including grasping and cutting, vacuum suction and plucking, twisting and pulling, and shaking and collecting. With this in mind, a proof of concept autonomous cotton robot was developed using a single-row harvesting module with a series of fingers and a cotton stripping mechanism attached to the front of the Amiga platform. This design effectively pulls the plant and harvests the bolls, which are then transported to a bucket in the center of the platform using two hoses and suction motors attached to the harvesting module. The retraction and extension of the harvesting modules are controlled by two electric cylinders, allowing for selective harvesting. Ultimately, this proposed design could serve as the basis for developing an efficient and autonomous cotton harvesting robot that could increase productivity.

**Author Contributions:** Conceptualization, M.F.M. and J.M.M.; methodology, M.F.M. and J.M.M.; software, n/a; validation, M.F.M., J.M.M., M.C., M.M., G.M. and E.B.; formal analysis, M.F.M.; investigation, M.F.M.; resources, J.M.M.; data curation, M.F.M.; writing—original draft preparation, M.F.M.; writing—review and editing, M.F.M. and J.M.M.; visualization, n/a; supervision, J.M.M. and E.B.; project administration, J.M.M. and E.B.; funding acquisition, J.M.M. and E.B. All authors have read and agreed to the published version of the manuscript.

**Funding:** This work was partially supported by a grant from Cotton Inc. Project No. 17-209 and is based on work supported by NIFA/USDA under project number SC-1700611. The authors express their gratitude to Clemson University Professional Internship and Co-op Program (UPIC). Student engagement is one of Clemson University's areas of investment. To meet this goal, Clemson developed an on-campus internship and co-op program (UPIC) in 2012 to offer students the opportunity to work closely with a member or members of Clemson's faculty or administration in an on-campus or Clemson-affiliated position.

**Institutional Review Board Statement:** Not applicable.

**Informed Consent Statement:** Not applicable.

**Data Availability Statement:** Not applicable.

**Conflicts of Interest:** The authors declare no conflict of interest.

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
