# Peer review of "Agricultural Harvesting Robot Concept Design and System Components: A Review"

_agriengineering, doi:10.3390/agriengineering5020048_

Round 1

Reviewer 1 Report

The paper provides an overview of the most recent harvesting robots' different concept designs and system components. The structure of the article is logical and investigates the research effort, developments and innovation in agricultural robots for field operations, and the associated concepts, principles and limitations.

In my opinion, the article will be better appreciated by the readers if the following points will be improved:

- the title suggests that the article is a review; however, if the authors want to bring a new concept about the development of a cotton harvesting robot they must offer more details regarding the proposed prototype;

- the paper will be much more explicit for readers if for each field/segment/section of the paper pictures will appear in addition to the summary tables.

- provide a subsection about limitations and gaps in the  harvesting robot field.

Author Response

We, the authors, appreciate all the time spent by the reviewers and all the ideas, suggestions, and comments presented. We addressed all the ideas, suggestions, and comments below in our revised manuscripts.

Comments and Suggestions:

  1. The title suggests that the article is a review; however, if the authors want to bring a new concept about the development of a cotton harvesting robot they must offer more details regarding the proposed prototype;

Response: The main objective of this review was to evaluate existing methods and concepts that could be used to develop an autonomous cotton harvesting robot. As there were only a few of research in cotton harvesting, we used other harvesting platforms to see if there might be similarities that we can learn, albeit the review. With the suggestion of the reviewer, we have updated the last subsection of the discussion, where we summarized the proposed new concept based on the review.

  1. The paper will be much more explicit for readers if, for each field/segment/section of the paper, pictures will appear in addition to the summary tables.

Response: We agree with the suggestion. We emailed the previous studies' main authors requesting permission to reprint the images in our manuscript and asking for photos at 300 dpi or higher. As suggested by the reviewer, we have included the images after the summary tables.

  1. Provide a subsection about limitations and gaps in the harvesting robot field.

Response: We added the subsection about limitations and gaps in the field of harvesting robots to our journal as suggested by the reviewer, which can be found on Lines 673~698.

4.6 Limitations and gaps in the field of harvesting robots

The development of harvesting robots has come a long way, but there are still many obstacles and gaps in technology. Some of the major limitations and gaps are:

  1. Crop variability: Harvesting robots need to be able to recognize and select different types of crops in different situations, such as different maturity levels and sizes. However, given the diversity of crops, this can be a difficult task, especially when dealing with complicated and delicate fruits and vegetables.
  2. Complicated environments: Farms often have complex environments with a wide range of topography, lighting, and weather. These difficult environments require harvesting robots to function, which can be a significant barrier to their growth.
  3. Cost: Harvest robots are currently quite expensive to develop and produce, which could discourage many farmers from using them. This could discourage smaller farmers from using harvesting robots.
  4. Restricted crop varieties: currently, harvesting robots are best suited for uniform and easy-to-harvest crops such as strawberries and lettuce. However, they may not be suitable for crops that require more specific harvesting methods.
  5. Safety issues: the development of harvesting robots must consider safety for both the crops being harvested and the workers who are near them. Their success depends on being able to work safely in the agricultural environment.
  6. Social acceptance: some farmers and customers may be opposed to the use of harvesting robots because they are concerned about how it will affect their jobs and conventional farming practices.

Addressing these limitations and shortcomings will be critical to the continued development and use of harvesting robots in agriculture. Some of these limitations will likely be addressed as technology evolves, but this will take time and continued investment in research and development.

Reviewer 2 Report

This paper presents a comprehensive review of the most recent harvesting robots. First, the authors introduced the general design concepts for these types of robots; then they analyzed different architectures for the system components. Finally, they discussed the main challenges and opportunities of the current state of the art. 

Two major issues have been found during the review:

- From the paper's title, it can be assumed that the analysis considered a wide variety of crops, however in the abstract (line 27), the introduction (lines 107 and 108), and the discussion section (line 551) indicate that this work focuses in the cotton harvesting. So, it is unclear if the presented review is for cotton only or other crops mentioned in the same work, such as tomato, strawberry, lettuce, etc. If this work focuses on the cotton harvesting process, a section explaining the specific characteristic of this crop should be included.

- In Section 3 (System Components) describes the general architecture of elements such as "Mobile platform," "Manipulators and effectors," "Sensing and localization," and "Path planning and navigation." It is very recommendable to include drawings or diagrams showing those architectures and the specific variations they typically present. 

Author Response

We, the authors, appreciate all the time spent by the reviewers and all the ideas, suggestions, and comments presented. We addressed all the ideas, suggestions, and comments below in our revised manuscripts.

Comments and Suggestions:

  1. From the paper's title, it can be assumed that the analysis considered a wide variety of crops, however in the abstract (line 27), the introduction (lines 107 and 108), and the discussion section (line 551) indicate that this work focuses in the cotton harvesting. So, it is unclear if the presented review is for cotton only or other crops mentioned in the same work, such as tomato, strawberry, lettuce, etc. If this work focuses on the cotton harvesting process, a section explaining the specific characteristic of this crop should be included.

Response: The main objective of this manuscript is to evaluate existing methods and concepts of agricultural harvesting robots for a variety of crops that could be used to develop an autonomous cotton harvesting robot. One of the issues with focusing only on cotton harvesting robots is the limited work focus on the development of the cotton harvesting robot. The main motivation is to review other crop harvesting robots so that we might be able to adapt some of their concepts to develop a mechanism for a cotton harvesting robot. As suggested, we have added the following sentences below in the discussion section (Lines 587 to 591) regarding a specific characteristic of cotton.

The cotton plant is characterized by being grown as a fiber crop for its soft, fluffy, and fibrous bolls. Cotton plants are generally grown in warm, humid climates and require a lot of water for growth. The cotton bolls are harvested and processed into cotton fiber, which is used to make a variety of textile products such as clothing, bedding, towels, and more.

  1. In Section 3 (System Components) describes the general architecture of elements such as "Mobile platform," "Manipulators and effectors," "Sensing and localization," and "Path planning and navigation." It is very recommendable to include drawings or diagrams showing those architectures and the specific variations they typically present.

Response: We agree with the recommendation. We were able to include images after the summary tables, as suggested by the reviewer for the Mobile Platform and Sensors and Localization subsections.

Round 2

Reviewer 1 Report

The manuscript has been significantly improved.

Reviewer 2 Report

Thank you for your detailed responses.  I am satisfied with your revision.